# Measurement of fasted state gastric antral motility before and after a standard bioavailability and bioequivalence 240 mL drink of water: Validation of MRI method against concomitant perfused manometry in healthy participants

Khaled Heissam[1]☯, Nichola Abrehart[1]☯, Caroline L. Hoad[1,2], Jeff Wright[1], Alex Menys[3], Kathryn Murray[1,2], Paul M. Glover[2], Geoffrey Hebbard[4], Penny A. Gowland[2], Jason Baker[5], William L. Hasler[5], Robin C. Spiller[1], Maura Corsetti[1], James G. Brasseur[6], Bart Hens[7,8], Kerby Shedden[9], Joseph Dickens[9], Deanna M. Mudie[7,10], Greg E. Amidon[7], Gordon L. Amidon[7], Luca Marciani[1]*

1 Nottingham Digestive Diseases Centre and NIHR Nottingham Biomedical Research Centre, University of Nottingham, Nottingham, United Kingdom, 2 Sir Peter Mansfield Imaging Centre, University of Nottingham, Nottingham, United Kingdom, 3 Motilent Ltd, London, United Kingdom, 4 Gastroenterology and Hepatology, The Royal Melbourne Hospital, University of Melbourne, Melbourne, Australia, 5 Division of Gastroenterology, University of Michigan, Ann Arbor, Michigan, United States of America, 6 Aerospace Engineering Sciences, University of Colorado, Boulder, Colorado, United States of America, 7 College of Pharmacy, University of Michigan, Ann Arbor, Michigan, United States of America, 8 Department of Pharmaceutical and Pharmacological Sciences, KU Leuven, Leuven, Belgium, 9 Department of Statistics, University of Michigan, Ann Arbor, Michigan, United States of America, 10 Capsugel, Bend, Oregon, United States of America

☯ These authors contributed equally to this work.

* Luca.Marciani@nottingham.ac.uk

**Data Availability Statement:** The anonymized individual data and the time courses data are available from the Nottingham Research Data

## Abstract

### Objective

The gastrointestinal environment in which drug products need to disintegrate before the drug can dissolve and be absorbed has not been studied in detail due to limitations, especially invasiveness of existing techniques. Minimal *in vivo* data is available on undisturbed gastrointestinal motility to improve relevance of predictive dissolution models and *in silico* tools such as physiologically-based pharmacokinetic models. Recent advances in magnetic resonance imaging methods could provide novel data and insights that can be used as a reference to validate and, if necessary, optimize these models. The conventional method for measuring gastrointestinal motility is via a manometric technique involving intubation. Nevertheless, it is feasible to measure gastrointestinal motility with magnetic resonance imaging. The aim of this study was is to develop and validate a magnetic resonance imaging method using the most recent semi-automated analysis method against concomitant perfused manometry method.

Management Repository (https://rdmc.nottingham.ac.uk/) with DOI 10.17639/nott.7068.

**Funding:** The authors GEA, GLA and LM received for this work a grant from the U.S. Food and Drug Administration (FDA), https://www.fda.gov/home, Contract HHSF223201510157C. This article therefore reflects the views of the authors and should not be construed to represent FDA's views or policies. The funders had no role in study design, data collection and analysis, decision to publish, or preparation of the manuscript.

**Competing interests:** I have read the journal's policy and the authors of this manuscript have the following competing interests: the author AM is the CEO of Motilent Limited.

**Abbreviations:** AC, ascending colon; AUC, area under curve; BA/BE, bioavailability/bioequivalent; CMP, comprehensive metabolic panel; CBC, complete blood count; DRAM, dual registration of abdominal motion; GB, gall bladder; GI, Gastrointestinal; MMS, Medical Measurement Systems; MRI, magnetic resonance imaging; PVC, Polyvinyl Chloride; SD, Standard Deviation; SPMIC, Sir Peter Mansfield Magnetic Imaging Centre; STMM, Spatio-Temporal Motility MRI; TC, transverse colon.

## Material and methods

Eighteen healthy fasted participants were recruited for this study. The participants were intubated with a water-perfused manometry catheter. Subsequently, stomach motility was assessed by cine-MRI acquired at intervals, of 3.5min sets, at coronal oblique planes through the abdomen and by simultaneous water perfused manometry, before and after administration of a standard bioavailability / bioequivalence 8 ounces (~240mL) drink of water. The magnetic resonance imaging motility images were analysed using Spatio-Temporal Motility analysis STMM techniques. The area under the curve of the gastric motility contractions was calculated for each set and compared between techniques. The study visit was then repeated one week later.

## Results

Data from 15 participants was analysed. There was a good correlation between the MRI antral motility plots area under the curve and corresponding perfused manometry motility area under the curve (r = 0.860) during both antral contractions and quiescence.

## Conclusion

Non-invasive dynamic magnetic resonance imaging of gastric antral motility coupled with recently developed, semi-automated magnetic resonance imaging data processing techniques correlated well with simultaneous, 'gold standard' water perfused manometry. This will be particularly helpful for research purposes related to oral absorption where the absorption of a drug is highly depending on the underlying gastrointestinal processes such as gastric emptying, gastrointestinal motility and availability of residual fluid volumes.

## Clinical trial

This trial was registered at ClinicalTrials.gov as NCT03191045.

## Introduction

The gastrointestinal (GI) environment in which drug products need to disintegrate prior to dissolution and absorption of the drug has not been studied in detail due to limitations, especially invasiveness of existing techniques. Minimal *in vivo* data is available on undisturbed GI motility. This information would be extremely helpful to improve the relevance of predictive dissolution models and computational software tools that are frequently applied in preclinical drug development to predict the systemic outcome of a drug. Classic studies of GI transit time of pharmaceutical dosage forms as a function of GI motility showed substantial gastric variability as a function of dosage form size and (unmeasured) motility state of the stomach [1]. Large variability of gastric emptying as a function of fasted motility state (phase, I, II, III) has been shown for fluids [2, 3] and particles [4–7]. Recent advances in magnetic resonance imaging (MRI) methods could provide novel data and insights [8–11]. On-going studies using MRI techniques have already shown that GI fluid (water) volumes can be measured [8, 11] though the dynamic motility environment remains to be studied in more detail, particularly in an undisturbed GI tract.

Intraluminal pressure recording (manometry) is considered the gold standard method to measure the contractile activity of the GI tract [12, 13]. This technique is, however, invasive and at times poorly tolerated by patients and this may disturb motility patterns. Parts of the bowel are inaccessible to conventional perfused manometry catheters, which are best suited to study the more proximal and distal part of the GI tract only. Moreover, the method may have low sensitivity to contractions that do not occlude the lumen [14–17].

Magnetic resonance imaging (MRI) provides an important potential alternative to manometry with some key advantages; these include the ability to image the entire bowel, acquire cross-sectional images, and the non-invasive nature of the technique. A number of recent publications describe MRI assessment of GI motility, including increased automation of analysis and quantification of GI motility biomarkers [18–23]. MRI particularly could address difficulties encountered with other techniques, such as collecting measurements in the presence of food or in children and frailer patients [15].

Dynamic MRI has been used to measure the frequency and amplitude of antral contractions in previous work [19, 20]. De Zwart *et al.* compared the use of MRI and the barostat to evaluate gastric motility and emptying disorders [24]. They assessed gastric accommodation to liquid and solid meals, at rest and also after infusion of glucagon and erythromycin, which alter gastric volume and motility respectively. The data showed that these methods significantly correlated with each other [21, 22]. It has also been shown that MRI can assess gastric motion and quantify the effects of metoclopramide and scopolamine on gastric motility [23]. Furthermore, the use of MRI to assess gastric motility and emptying has been validated against gamma scintigraphy [25–28].

Over the last two decades, MRI has been used to provide quantitative detailed information on gastric motor function. An early study compared MRI assessment of antro-duodenal motility with simultaneous water perfused manometry [29] and noted that perfused manometry under-detected gastric contractions compared to MRI. In another study, a combination of MRI and manometry was used to investigate antropyloroduodenal motor activity and its relation to gastric emptying [24, 30]. These results showed that manometry missed approximately 20% of the contraction waves which were detected by MRI [24]. Neither of the studies used semi-automated analysis methods.

Recently, a new method of making semi-automated, Spatio-Temporal Motility MRI (STMM) measurements of gastric and colonic contractions, based on changes in bowel lumen diameter, has been introduced [31]. This has the potential to speed up data analysis and make it less operator dependent. However, the combination of MRI and STMM analysis has not yet been tested *in vivo* against simultaneous conventional manometry.

In this study, we aimed to validate automated MRI motility measures using the new STMM mapping technique [31] against simultaneous water perfused manometry in healthy adult participants.

## Materials and methods

### Participants

The study was conducted at the Sir Peter Mansfield Imaging Centre (SPMIC) at the University of Nottingham. The study was approved by the University of Nottingham Faculty of Medicine and Health Sciences Research Ethics Committee (A14112016) and by the US Food and Drug Administration Research Involving Human Participants Committee (16-073D). All participants gave written, informed consent before joining the study.

For inclusion, the participants had to be healthy and aged 18–60 years old. Health checks included a medical history questionnaire, blood pressure and arterial pulse check. A 10 mL

venous blood sample was taken from the forearm for a comprehensive metabolic panel (CMP) and complete blood count (CBC) analysis. Exclusion criteria encompassed use of any medication which interferes with gastrointestinal motility, working night shifts, strenuous exercise for > 10 h/week, known alcohol dependency, contraindications for MRI scanning as assessed by a standard MRI safety questionnaire, pregnancy and inability to lie flat.

Eighteen healthy volunteers were recruited (9 males and 9 females). The mean age was 29 years old (with a standard deviation (SD) of 10, ranging from 19 (minimum) to 55 (maximum) years old) and the mean body mass index was $24 \pm 2$ kg/m$^2$. They took part in two identical fasted state MRI study visits with a minimum of 1 week between visits. The study was registered with ClinicalTrials.gov with identifier NCT03191045.

## Experimental design

This was a single-centre, open-label design study that consisted of two separate identical study days up to 4 weeks apart. Participants consumed their habitual diet between each visit. Each study visit lasted approximately 8 hours and it was divided into 5 sequential parts as shown in Fig 1 with a focus on the fasting state periods of motility, as the fasting state is of particular interest for oral solid drug delivery products. On arrival, the participants completed a study day eligibility questionnaire to confirm eligibility and document adherence to overnight fasting.

The study day began with a naso-duodenal catheter intubation which took place in a seated position. A local anaesthetic (Xylocaine® spray, AstraZeneca Ltd, UK) and a small amount of water-soluble lubricant were used to ease passage through the nose (Optilube 5 g sachets, Optimum Medical Solutions, Leeds UK). The naso-duodenal catheter was a custom design by Mui Scientific (Mississauga, ON, Canada) made of clear polyvinyl chloride (PVC) plastic extrusion tubing. It was a 16 channel, single-use, MRI compatible, water perfused catheter with an external diameter of 12F (4.0 mm) and luminal diameter of 0.3 mm and a core of 1.0 mm with side holes for pressure measurements placed at 5 cm intervals.

The perfused manometry system pressure to drive the water through the catheter was set at 1 bar resulting in a catheter perfusion rate of 1 mL/minute. The total length of the tube was 180 cm plus an additional 100 cm of pigtails. The tip of the tube had a small non-latex balloon attached to allow for inflation with 5–10 mL water for MRI localization. After intubation and resting, the participants were positioned in the MRI scanner. The latex balloon was inflated to determine that the catheter had passed the pylorus and was positioned correctly with recording ports in the stomach. The perfused manometry ports tend to record primarily antral contractions whilst more proximal sensors in the wider fundal region do not record contractions well, as such the assessment here was focused on the antral region of the stomach.

The balloon was then deflated and after a short period of adaptation, the catheter was connected to the MRI compatible water-perfused manometry system (Biomedical Engineering

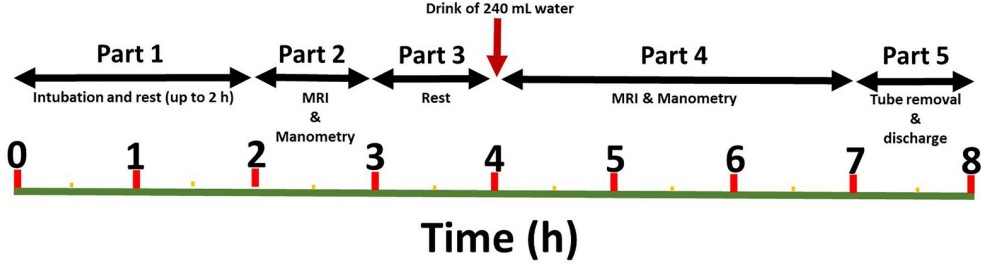

**Fig 1. Study diagram.** Schematic diagram of the study day timeline.

Department, The Royal Melbourne Hospital, Melbourne, Australia). The pressure was converted to electrical signal just outside the scanner bore using transducers connected to the pigtails and secured to a MRI-compatible trolley placed next to the scanner bed. Electric cables and pipes went through a wave guide opening and connected the transducers box to the electronics and pressure gas cylinder placed outside the scanner room. To set reference pressure for the catheter ports, the manometry machine was calibrated manually by gravity immediately before and after every study session and the calibration recorded on the study session's traces so that malfunction or drift could be detected. An acquisition sample frequency of 25 Hz was used. Each of the differential input lines to the pressure amplifier unit was decoupled using 470 pF capacitors to the reference ground of the amplifier. This effectively trapped otherwise obtrusive 64 MHz RF pulses picked up from the scanner without affecting low-frequency motility information. The pressure data was stored in a digital format data logger (Trace 1.3, Biomedical Engineering Department, The Royal Melbourne Hospital, Melbourne, Australia). At the end of each manometric study, the data log was extracted to be subsequently analysed using common commercial software (MMS, Medical Measurement Systems B.V., Enschede, The Netherlands).

The second part of the study day consisted of one hour of baseline MRI measurements with simultaneous water-perfused manometry data acquisition. MRI scanning was carried out supine using a 1.5T General Electric HDX MRI scanner (General Electric Healthcare, Little Chalfont, Buckinghamshire, UK). Stomach motility was assessed at intervals using a cine-MRI acquisition set at coronal oblique planes through the abdomen, aligned to include the antral region of the stomach. Each dynamic motility MRI scan acquired 60 sets of 4 slices over a time window of approximately 3.5 min. Each dynamic scanning set was acquired across the abdomen whilst the participant breathed gently. The data were acquired using a FIESTA (True-FISP) sequence (TR 3, TE 0.9, Flip angle 45, Slice thickness 8 mm, field of view 40 mm, matrix size 256, and pixels 1.56 mm × 1.56 mm × 8 mm) at a repetition time for a given plane of 3.7 sec for a total duration of 224 sec.

During the third part of the study day, the participants were allowed to rest for up to an hour outside the scanner room (during this period the manometry system was disconnected). None of the participants wanted a long break, they took only advantage of the rest time to stretch their legs and visit the toilet before continuing and the average time including reconnection was 15 min. Afterwards, they returned back to the scanner, the perfused manometry system was reconnected and the participants drank a standard water challenge of 240 mL while they were sitting up on the scanner bed. They were not instructed to drink the water at a given speed, but all of the participants drank the glass of water quickly, within approximately one minute. A drink of 240 mL of water is the current volume of water recommended by the US Food and Drug Administration for bioavailability/bioequivalence (BA/BE) studies in the fasted state [32, 33]. For the fourth part of the study, the participants laid in the scanner for up to 3 hours of MRI and manometry data were collected concurrently with an optional 20 minutes comfort break towards the end. Throughout each part of the study, the cine-MRI acquisition blocks were interleaved with T2-weighted MRI scans carried out to visualise the state of fluid volumes in the abdomen and also catheter positioning by inflating temporarily the tip balloon with a small volume of fluid.

After the data collection period, the participants moved to the clinical room to remove the tube, they were provided with refreshments and were then discharged.

## Data analysis

**Magnetic resonance imaging (MRI).** The time series 'cine' data were corrected for respiratory motion using Dual Registration of Abdominal Motion (DRAM) GIQuant (Motilent,

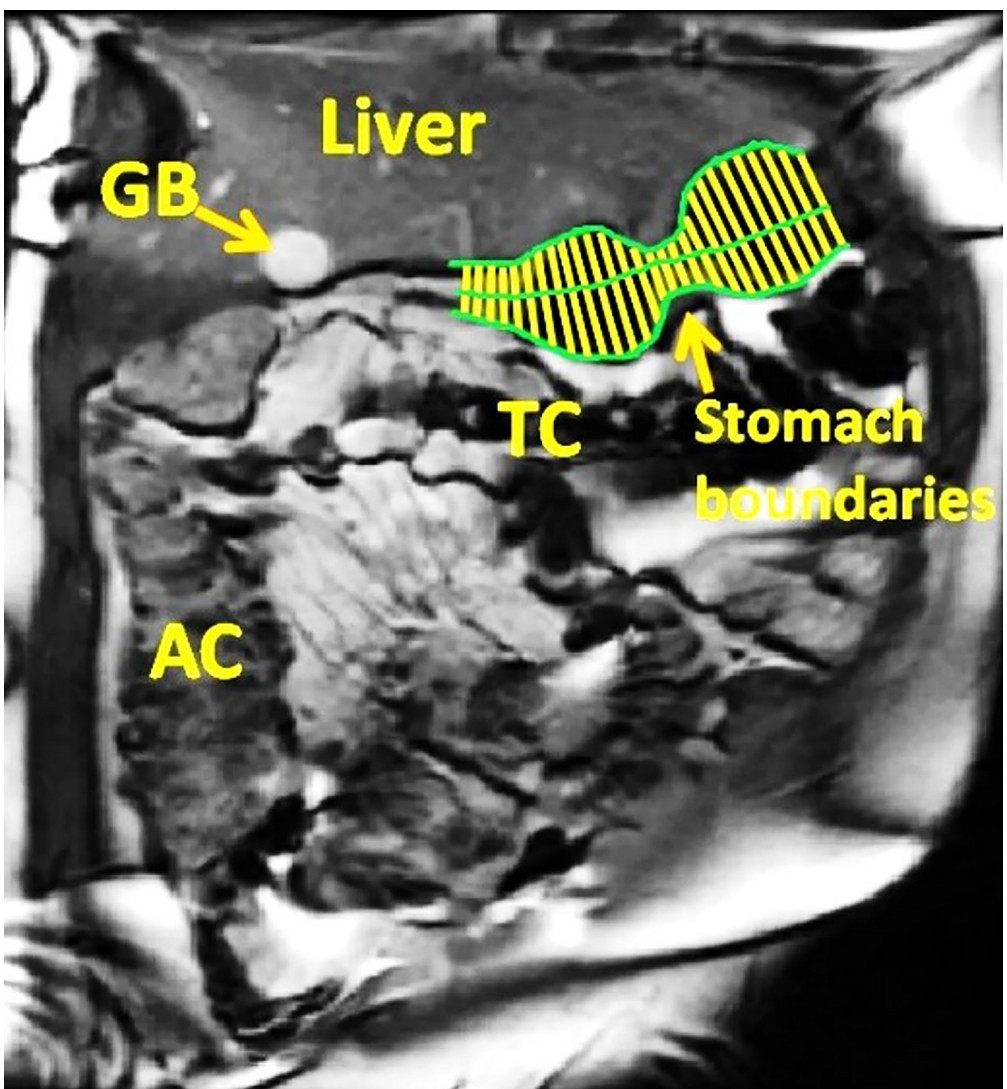

**Fig 2. Image analysis.** Example of MRI data analysis. The stomach wall boundaries and the axis of the stomach are shown in green. The yellow lines represent the lumen diameter perpendicular to the stomach axis. AC, ascending colon; TC, transverse colon; GB, gall bladder.

London, UK) to provide information on the GI tract wall motion which allowed subsequent automatic image analysis [34]. The data were then analysed using a Spatio-Temporal Motility MRI technique (GIQuant, Motilent, London, UK) as recently described [31]. Briefly, the stomach wall boundaries were first identified manually on one of the images. The boundaries were then propagated automatically through the data set to collect wall boundary coordinates on all images. The axis of the stomach was then drawn manually along the visible part of the lumen as indicated by the green lines in Fig 2. Once done, the software automatically measured the lumen diameter as a function of time at defined node points along the stomach axis as shown by the yellow lines perpendicular to the stomach axis in Fig 2. This was performed for each time point so that plots of the stomach diameter versus time were generated. The area under the curve (AUC) was then calculated over the entire 3.5 min acquisition block. The output was AUC in mm × second [34].

**Water-perfused manometry.** The water perfused manometry data were analysed using commercial MMS software (Laborie, Mississauga, ON). Respiratory artefacts were removed. The manometry traces from the gastric ports were segmented to synchronise with each 3.5 min MRI acquisition. The output was the AUC in mmHg × second [35].

## Statistical analysis

The study design included power analysis considerations. Briefly, for correlation analyses involving data that are summarized to one number per subject, the standard error for the estimated correlation coefficient over 15 subjects is 0.28. However, our main analyses use time-resolved data with an overall sample size of 421 values over the 15 subjects. Using a Kronecker sum model to capture within-modality autocorrelation (for MRI or manometry) and between-modality correlation (the parameter of interest), much smaller standard errors are obtained. For example, if the within-modality autocorrelation is 0.5, the between-modality correlation coefficient is estimated with a standard error of 0.049. Thus, the confidence intervals for the correlation parameter of interest will be reported as the estimate +/- 0.1, giving us excellent power to resolve weak from strong correlation. The standard error depends on the within-modality autocorrelation, which was not known at the time of study design. The actual data obtained showed high precision of the estimated correlation coefficient, indicating that the a priori study design was successful.

Data were analysed using SPSS for Windows (Release 24.0, Chicago, IL) and Graph Pad Prism 7 (GraphPad Software, San Diego, CA). All data are presented as mean ± standard deviation unless otherwise indicated. A Pearson's correlation was used to assess the agreement between MRI area under the curve and corresponding manometry area under the curve. In addition, mixed-effects regression was used to assess the relationship between MRI and manometry measurements of gastric antral motility, while accounting for repeated measurements within study participant and study visits. The conditional mean of the MRI 3.5 min AUC was estimated as both a linear and nonlinear function of the manometry 3.5 min corresponding AUC. There was some evidence of a nonlinear association between manometry and MRI 3.5 min AUC measurements. A natural spline basis was used to assess whether the data support a non-linear conditional mean structure. The mixed effects models included random intercepts for each subject and for each visit-within-subject, and random slopes for each participant.

## Results

The study procedures were well tolerated and there were no adverse events. All the eighteen participants returned for their second visit. Data from three participants had to be excluded because of technical difficulties with either the MRI, the perfused manometry hardware or the intubations, leaving 15 participants for analysis.

The MRI motility plots showed good visual correspondence compared to the perfused manometry traces during both antral contractions and quiescence. Three examples of 3.5 min acquisition blocks are shown in Fig 3. The two top panels show gastric contraction peaks detected synchronously by MRI and perfused manometry, whilst the bottom panel shows a period of quiescence sampled by both techniques. The perfused manometry traces allowed standard quality monitoring of the antral motility in the participants as described in the materials and methods section.

Fig 4 shows the average AUC of the MRI stomach antral motility for each participant plotted against the corresponding average AUC of the water perfused manometry (n = 15). As

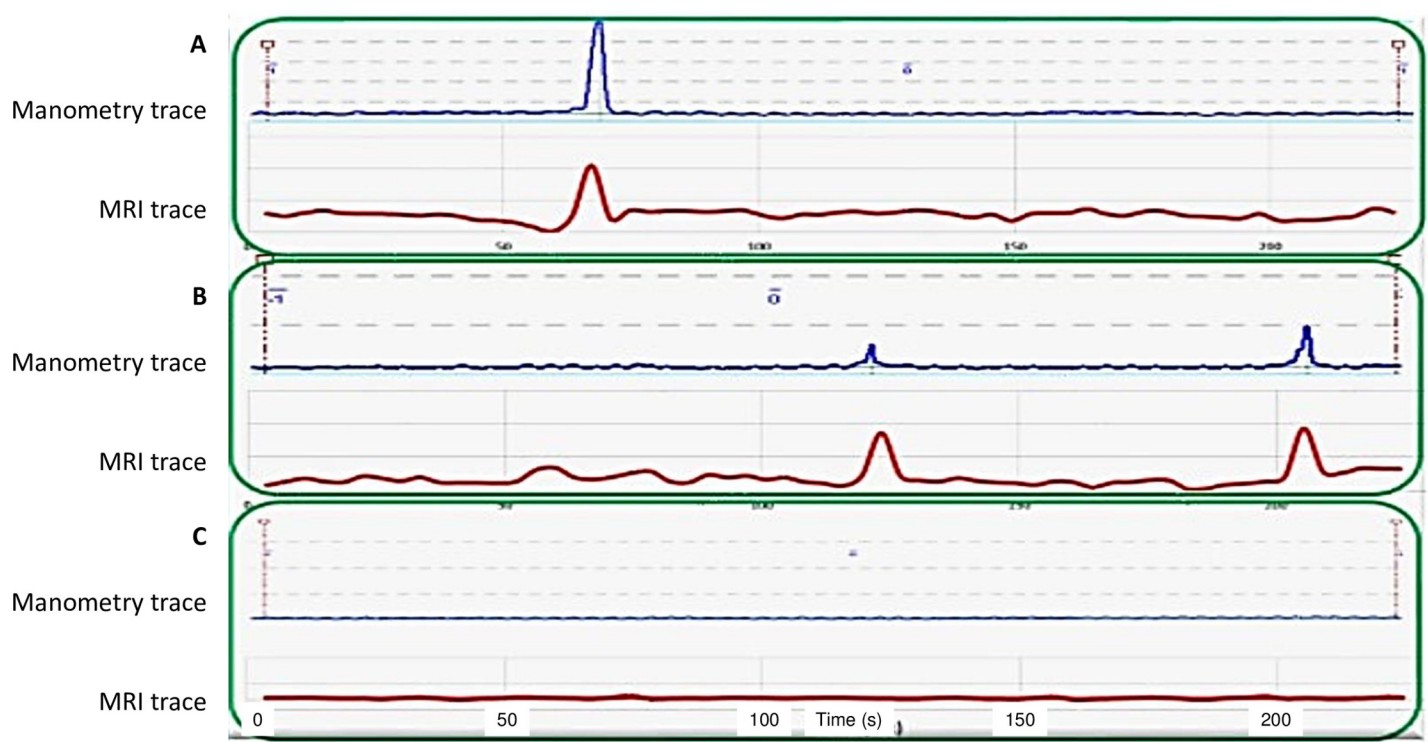

**Fig 3. MRI and manometry plots.** Examples of MRI gastric antral motility plots (in red) and corresponding perfused manometry traces (in blue) from different participants, during antral contractions (A and B) and quiescence (C).

indicated by the Pearson's correlation coefficient, a strong and significant correlation was observed between both data sets (r = 0.860).

Having considered the average AUCs as above, it is also important to consider all individual data points of the MRI motility AUCs plotted against the corresponding perfused manometry

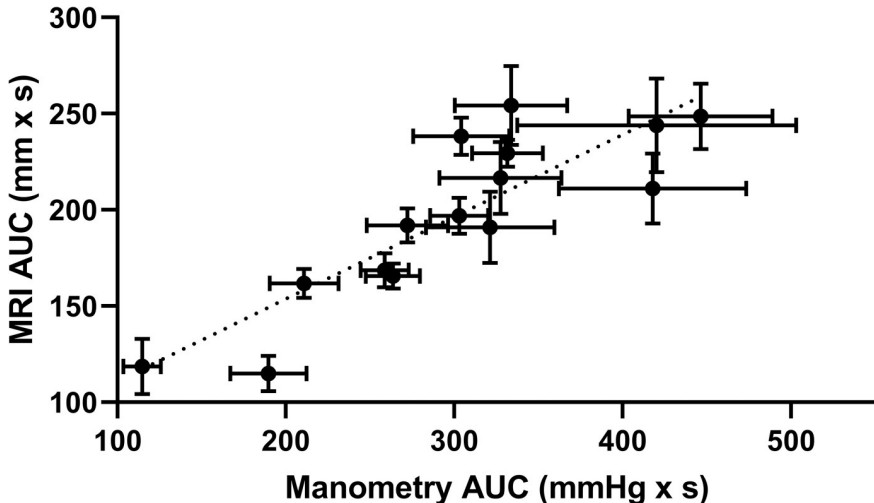

**Fig 4. Comparison of MRI and manometry mean area under the curves.** Mean (± SEM) values of manometry AUC for each participant plotted against the corresponding mean MRI AUC (n = 15) during one of the visits (Pearson's correlation coefficient, r = 0.860).

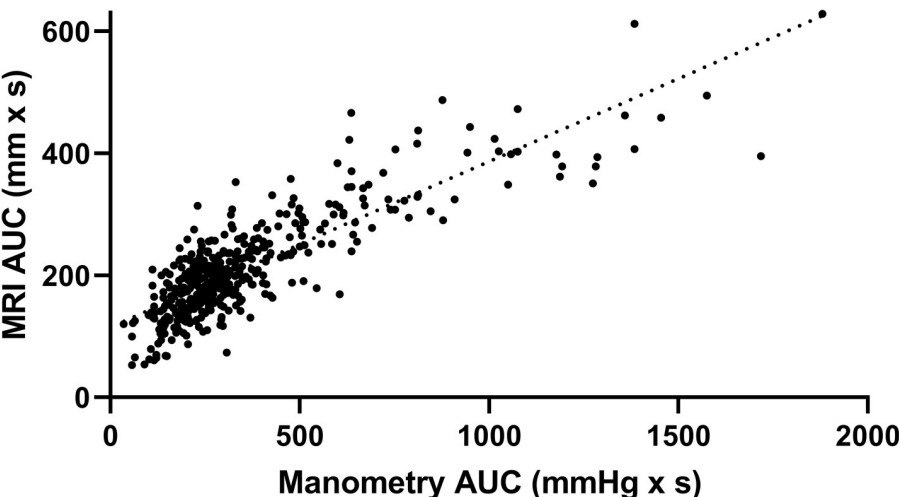

**Fig 5. Comparison of MRI and manometry individual area under the curves.** Scatter plot of individual values of MRI antral motility for each 3.5 min imaging block AUC and the corresponding 3.5 min perfused manometry motility AUC for all participants, thereby providing n = 421 data points (Pearson's correlation coefficient, r = 0.843).

motility AUCs across the entire study (n = 421, Fig 5). These individual points correlate directly each individual trace without averaging noise from either measurement system. A positive, strong correlation was observed between the two techniques (r = 0.843).

The association between MRI and manometry measurements of gastric antral motility was strong. Evidence of moderate between subject differences in MRI AUC, after controlling for manometry AUC, was found. Controlling for manometry AUC and study subject, substantial between-visit variation in MRI AUC was not observed. This suggests that there exist stable, unobserved subject-level characteristics that affect MRI AUC after accounting for manometry AUC. In addition, we also assessed whether the association between MRI and manometry AUC measures is non-linear. The non-linear model described the observed data significantly better compared to the linear model ($\chi_3^2 = 57.2$, p < 0.001).

The association between MRI and manometry measurements was not found to vary substantially between subjects. The fitted non-linear model estimates that one standard deviation increase in manometry AUC is associated with an MRI AUC increase of 100 mm x min for the average subject, when manometry AUC is within one standard deviation of its mean value. For a single subject, the model predicts that his or her expected MRI AUC increase will be within 10 mm x min of the population average increase of 100 mm x min. The non-linear model described the AUC data better than the linear model.

The AUC values can also be aggregated for all subjects at consecutive time points during the study to show the time courses of gastric antral motility during the study for both MRI and perfused manometry. These time courses are plotted together in Fig 6 to show visually the good correspondence between the two time courses.

## Discussion

The main objective of this study was to validate MRI motility measures automated using new semi-automated MRI image analysis techniques against simultaneous water perfused manometry in healthy adult participants. The fasting state and the use of a 240 mL dose of water as a challenge were specifically chosen because of their relevance to the bioavailability/bioequivalence (BA/BE) studies in the fasted state [32, 33]. However, it should be noted that the participants here did not

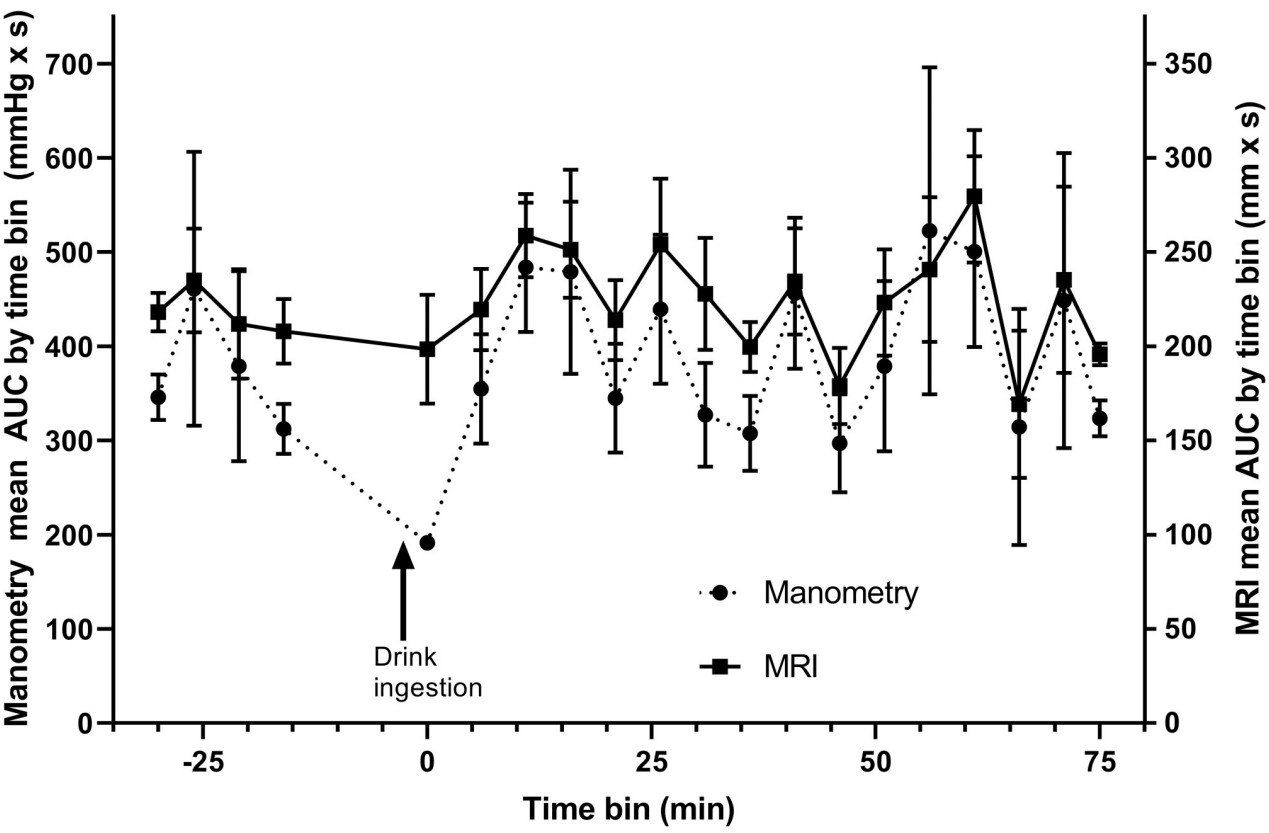

**Fig 6. Time courses of gastric antral motility.** Time courses of the area under the curve of gastric antral contractile activity observed with MRI (solid line) and water-perfused manometry (dotted line) aggregated for all subjects at consecutive intervals (mean ± SD).

actually ingest a drug product. The recommended 240 mL of water was applied as this volume is also used in BA/BE studies during the clinical phase of drug product development. A water drink challenge has indeed been shown to stimulate gastric motility [36].

The study clearly demonstrated a strong correlation between the two methods. The association between MRI AUC and manometry AUC was found to be rather non-linear than linear. This may be due to the fact that MRI and manometry measure different features of gastric motility. The MRI AUC is a function of the distance between the opposing stomach walls, which cannot decrease below zero millimetres. In the case of manometry, pressure at the surface of the gut reflects the force applied to a small area element on the effective mucosal surface, by the fluid when the surface element is not in direct contact with other mucosal surface elements (intrabolus hydrodynamic pressure), or when the surface element is in direct contact with other mucosal surface elements (contact pressure) [37]. It is generally the case with peristaltic motility that when pressure exceeds 15–35 mmHg (referenced to mediastinal or abdominal pressure as appropriate), the manometric port is generally measuring mucosal contact pressure. Below those values, manometry generally measures hydrodynamic pressure at the interface with intrabolus fluid [37].

Therefore, manometry AUC is a function of the pressure, which can continue to increase while MRI measures a distance of zero mm between the gastric walls. When the stomach is active, this will cause the manometry AUC to increase more than the MRI AUC, inducing a non-linear relationship between the modalities. It should be noted that the non-linear model especially improved with respect to large manometry AUC values. In the lower range,

however, a clear linear regression was observed. Conversely, Fig 6 shows a large discrepancy between MRI and manometry AUCs with manometry being lower than MRI immediately after the 240 mL drink. This is likely to reflect the inability of the perfused manometry to measure non-occlusive events when a free fluid is present to redistribute the pressure changes instantaneously.

To directly quantify the details of the contractions of the gut, an imaging modality that can resolve time changes in lumen geometry is required. Imaging and manometry are, in fact, complementary since imaging identifies space-time changes in lumen geometry but does not quantify muscle squeeze, while manometry directly measures muscle squeeze but not the geometry of the lumen. Ideally, one wishes to measure both simultaneously, however, the technological challenges for concurrent measurement are sufficiently great that a single modality data collection—pressure or imaging—is the norm. Of great interest, therefore, are the potential relationships between manometric pressure data and imaging data during contractile activity along the gut [37] and interpretations in context with muscle function [38].

The contemporary dynamic fast acquisition of MRI has the advantage of being non-invasive, safe and uses non-ionising radiation, allowing assessment of stomach contractions with good spatiotemporal resolution [20, 39, 40]. Moreover, in comparison to manometry, the MRI images provide superb soft-tissue contrast, which aids direct assessment of the luminal occlusive and non-occlusive gastric contractions. At the same time, it allows the evaluation of the surrounding anatomical structures. MRI can not only assist in evaluating the response to food or a pharmacological agent but also help in a better understanding of the pathophysiology of gastric motility disorders and drug development [21].

Whilst MRI has been repeatedly proposed as an alternative to invasive manometry, only a few studies addressed the issue of validation of the MRI method, particularly with a view of assessing recent automated data processing methods like STMM. Previous studies were positive but were small scale studies that acquired data for relatively short time windows [24, 29]. There is also a feasibility study on colonic motility in which dynamic MRI has been compared to perfused manometry [39], which found a strong correlation between MRI visualized colonic movements and intraluminal pressure changes. Advances in MRI hardware including improved parallel imaging techniques have enabled rapid image acquisition over a large volume coverage, leading to reduced effects of motion artefacts due to respiration and peristalsis [17]. In addition, MRI can assess the proximal and distal stomach regions simultaneously which cannot be done by ultrasound [11, 26, 41, 42]. Also, the non-invasive characteristic of MRI opens perspectives towards screening gastric motility in different populations (*e.g.* elderly and paediatrics). Non-invasive monitoring of gastric motility has the potential to be performed in conjunction with MRI monitoring of other physiological parameters such as evaluation of the gastric emptying time and measurement of GI fluid volumes, which can have remarkable effects, for instance, on orally administered drug's residence time and dissolution, respectively [43]. The fact that numerous physiological variables can be studied simultaneously, provides a unique data set that can help, for instance, computer modelling of gastric function [44] and *in vitro* tools to be further developed and validated. With respect to future uses of the MRI technique, a pharmacokinetic study in combination with assessment of motility could provide unprecedented insights on inter-subject variability in systemic outcome of a drug, as the underlying motility patterns are likely one dominant cause of known inter-subject differences in formulation disintegration and, subsequently, dissolution and absorption of the drug. These type of studies are the next logical step to further our understanding of to investigate the actual assessment of pharmacokinetics variables.

Our study collected a large amount of MRI data which would have been very time consuming to analyse manually with individual drawings to measure changes in the gastric luminal

diameter over time. Much of the analysis was however done semi-automatically with the STMM technique, which saved considerable data processing time and was less operator dependent than a manual approach.

One limitation of the MRI technique is that it cannot continuously measure the motor response of the gut over long periods of time (hours) as ambulatory manometry does. This is not only because of cost and availability, but also because prolonged periods in the scanner bore can be uncomfortable for patients. Therefore, the MRI methods may be more suitable to assessing immediate impact of interventions such as meals and drugs over a relatively short time period. The planned analysis of the motility traces was focused here on the area under the curve of motility as primary outcomes of this study. The AUC incorporates both amplitude and frequency of contractions in a single integral. In future more advanced analysis parameters such as the Motility Index (a composite measure still incorporating both contractions frequency and amplitude and also representing the fraction of time during which gastric motility was detected) could be considered.

One may argue that the intubation of a catheter may disturb GI physiology. However, it should be noted that the impact of transpyloric tubes on gastric emptying has been explored in the eighties by Müller-Lissner and co-workers and no effect was observed [45]. Moreover, in this study we allowed the participants to rest (up to 2 h) before the start of the data collection. This gave the participant time to relax and get comfortable with the presence of the catheter. To the extent of our knowledge, it is not known how motility may be influenced by the position of a person (standing versus lying down). However, most of the motility studies performed with manometry are those where the subject/patient is lying in a supine position. A study performed by Treier and co-workers demonstrated differences in gastric emptying after eating a solid/liquid meal in a lying body position or seated position [46]. However, body position did not show any effect on gastric relaxation and initial gastric volumes. In another interesting study the same group placed participants upright or upside down in an open-design MRI scanner and concluded that the rate of gastric emptying was maintained despite the two opposite body positions [47].

Data from MRI studies will play a pivotal role in the validation of predictive in vitro and in silico tools as frequently used by formulation scientists in pharmaceutical drug development. For example, MRI data from a previous study were implemented in a physiologically-based pharmacokinetic (PBPK) simulation tool to adequately reflect the residual fluid volumes in the different compartments of the GI tract [48, 49]. The acquired data from this study can be further used to revise GI motility and transit times in these platforms to make predictions with better accuracy and precision.

## Conclusions

In conclusion, cine-MRI coupled with recently developed, semi-automated STMM data processing technique is a promising method to assess gastric antral motility, which produced results that were well correlated with simultaneous, 'gold standard' water perfused manometry. Dynamic MRI is non-invasive and provides unique data on the undisturbed GI environment. In the field of oral biopharmaceutics, the presented data are of importance to serve as a reference for validation of existing *in vitro* and *in silico* tools that are frequently applied to predict the *in vivo* performance of orally administered drug products.

## Acknowledgments

Bart Hens is a postdoctoral researcher of the Flemish Research Council (FWO - 12R2119N).

## Author Contributions

**Conceptualization:** Greg E. Amidon, Gordon L. Amidon, Luca Marciani.

**Data curation:** Khaled Heissam, Nichola Abrehart, Luca Marciani.

**Formal analysis:** Khaled Heissam, Nichola Abrehart, Caroline L. Hoad, Jeff Wright, Alex Menys, Kathryn Murray, Kerby Shedden, Joseph Dickens.

**Funding acquisition:** Greg E. Amidon, Gordon L. Amidon, Luca Marciani.

**Investigation:** Khaled Heissam, Nichola Abrehart.

**Methodology:** Nichola Abrehart, Caroline L. Hoad, Jeff Wright, Alex Menys, Paul M. Glover, Geoffrey Hebbard, Penny A. Gowland, Jason Baker, William L. Hasler, Robin C. Spiller, Maura Corsetti, James G. Brasseur, Bart Hens, Deanna M. Mudie, Luca Marciani.

**Project administration:** Greg E. Amidon, Gordon L. Amidon, Luca Marciani.

**Supervision:** Greg E. Amidon, Gordon L. Amidon, Luca Marciani.

**Writing – original draft:** Khaled Heissam, Nichola Abrehart, Luca Marciani.

**Writing – review & editing:** Khaled Heissam, Nichola Abrehart, Caroline L. Hoad, Jeff Wright, Alex Menys, Kathryn Murray, Paul M. Glover, Geoffrey Hebbard, Penny A. Gowland, Jason Baker, William L. Hasler, Robin C. Spiller, Maura Corsetti, James G. Brasseur, Bart Hens, Kerby Shedden, Joseph Dickens, Deanna M. Mudie, Greg E. Amidon, Gordon L. Amidon, Luca Marciani.

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
