## [Decision Letter · Decision Letter 0]

15 Jul 2020

PONE-D-20-12517

Measurement of fasted state gastric motility before and after a standard bioavailability and bioequivalence 240 mL drink of water: validation of spatio-temporal mapping MRI imaging method against concomitant perfused manometry in healthy participants

PLOS ONE

Dear Dr. Marciani,

Thank you for submitting your manuscript to PLOS ONE. After careful consideration, we feel that it has merit but does not fully meet PLOS ONE’s publication criteria as it currently stands. Therefore, we invite you to submit a revised version of the manuscript that addresses the points raised during the review process.

The reviewers had a number of concerns about the methodology and statistical analysis in the study. They require clarification about several of the procedures/methods used, and the use of correlation coefficients. The comments can be viewed in full below.

We look forward to receiving your revised manuscript.

Kind regards,

Natasha McDonald, PhD

Associate Editor

PLOS ONE

Journal Requirements:

"The study was approved by the local Research Ethics Committee (A14112016) and by the US Food and Drug Administration Research Involving Human Participants Committee (16-073D). All participants gave written, informed consent before joining the study."

4. Thank you for stating the following in the Financial Disclosure section:

' The authors GEA, GLA and LM received for this work a grant from the U.S. Food and Drug Administration (FDA), https://www.fda.gov/home, Contract HHSF223201510157C. The funders had no role in study design, data collection and analysis, decision to publish, or preparation of the manuscript. This article therefore reflects the views of the authors and should not be construed to represent FDA’s views or policies.'

We note that one or more of the authors are employed by a commercial company: Motilent Ltd, and Capsugel

a. Please provide an amended Funding Statement declaring these commercial affiliations, as well as a statement regarding the Role of Funders in your study. If the funding organization did not play a role in the study design, data collection and analysis, decision to publish, or preparation of the manuscript and only provided financial support in the form of authors' salaries and/or research materials, please review your statements relating to the author contributions, and ensure you have specifically and accurately indicated the role(s) that these authors had in your study. You can update author roles in the Author Contributions section of the online submission form.

b. Please also provide an updated Competing Interests Statement declaring these commercial affiliations along with any other relevant declarations relating to employment, consultancy, patents, products in development, or marketed products, etc.  

Within your Competing Interests Statement, please confirm that these commercial affiliations do not alter your adherence to all PLOS ONE policies on sharing data and materials by including the following statement: "This does not alter our adherence to  PLOS ONE policies on sharing data and materials.” (as detailed online in our guide for authors http://journals.plos.org/plosone/s/competing-interests) . If this adherence statement is not accurate and  there are restrictions on sharing of data and/or materials, please state these.

Please note that we cannot proceed with consideration of your article until this information has been declared.

Reviewers' comments:

Reviewer's Responses to Questions

**Comments to the Author**

1. Is the manuscript technically sound, and do the data support the conclusions?

Reviewer #1: Yes

Reviewer #2: Yes

Reviewer #3: Partly

2. Has the statistical analysis been performed appropriately and rigorously? 

Reviewer #1: No

Reviewer #2: Yes

Reviewer #3: Yes

3. Have the authors made all data underlying the findings in their manuscript fully available?

Reviewer #1: Yes

Reviewer #2: Yes

Reviewer #3: Yes

4. Is the manuscript presented in an intelligible fashion and written in standard English?

Reviewer #1: Yes

Reviewer #2: Yes

Reviewer #3: Yes

5. Review Comments to the Author

Reviewer #1: AUCs are compared using MRI for gastric motility. There is not much statistical content, other than some Pearson correlations and a mixed model. So I would argue about the term "spatio-temporal", as there is no multidimensional spatial/time series analyses, but only a simple repeated measures. Sample size is small and seems to have been chosen at random, as there is no power computation. P-values on correlations are really meaningless, because they test 0 correlation against "some" correlation. In this case, a significant p-value doesn't necessarily mean the correlations are strong. The regression models need a careful residual analysis. I think there could have been a lot more done with these data, but maybe the sample size is too small.

Reviewer #2: This manuscript reports on the use of non-invasive dynamic magnetic resonance imaging coupled with semi-automated spatio-temporal data analysis for measuring fasted-state gastric motility and validation of such a method vs. intraluminal pressure recording by means of perfused manometry technique.

Human gastrointestinal motility patterns are of utmost importance to oral drug delivery. The investigated RMI-based method would represent a valuable advancement in the relevant study and allow to gain better insight into the impact of many variables. The work is very interesting, broad in scope, clearly exposed and in-depth discussed.

Please find minor comments below.

- The title is rather long: is the word “imaging” actually needed when the acronym RMI already contains it?

- Lines 113-114: as the two methods correlated with each other, “both” sounds odd and can be deleted.

- Lines 123-124: “Both studies did not use semi-automated analysis methods” may better read “Neither of the studies used…”.

- Line 125: please delete the comma after “making”.

Lines 127-128: “more operator independent” may be changed into “less operator-dependent” (this also applies to line 400).

Line 143: “Exclusion criteria included..” sounds a bit strange. “Exclusion criteria were”? Or “encompassed”?

Fig. 2: the caption should be changed into a more concise form taking account of the detailed description and explanation already provided in the text. When the content of the image and the meaning of green and yellow lines are indicated, it will be enough.

Line 257: “Each subject participated in either one or two intubation studies” seems to clash with the Methods section reporting that the volunteers “…took part in two identical fasted state MRI study visits” (lines 149-150) and “This was a single-centre, open-label design study that consisted of two separate identical study days...” (lines 154-155).

Lines 297-298: please check and rephrase “The correlation for these data shows a significant positive correlation …”.

Fig. 6: please check the legend as colors are not visible in the figure whereas dotted/solid lines and different symbols are used as distinguishing features for the 2 curves.

Lines 337-340: please check and rephrase” The fasting state and the use of a 240 mL dose of water as a challenge were specifically chosen because of their relevance to the bioavailability/bioequivalence (BA/BE) studies in the fasted state”, as it is implied that fasting volunteers take part in fasted-state bioavailability/bioequivalence studies. It seems that only 240 ml of water was chosen because of relevance to those studies.

Lines 354-355: “When the stomach is active, this will cause the manometry AUC to increase more quickly than the MRI AUC….” does not appear strictly related to the previously-mentioned circumstance. Could this be an additional factor differing between the two methods?

Line 364: should “complimentary” read “complementary”?

Line 382: “a relative short time window” (or without “a”).

Line 393-394: “an oral administered” should read “an orally administered”

Line 396: as no statistical optimization study is concerned, “optimized” could be replaced by a different verb. The same applies to lines 418 and 438 (“optimization”).

Lines 315-319: this concept is already mentioned at lines 390-394. Please revise or synthesize the two similar passages in order to avoid repetitions.

Lines 431: “gastric motility” can be deleted as it is specified again at line 432, same sentence.

Reviewer #3: The manuscript is interesting and well written. It is not the first time that MRI has been shown to be a valid method for the measurement of gastric emptying, secretion, motility and intragastric distribution of gastric contents. This study, however, adds value to the use of MRI that assess gastric motility by using a new spatio-temporal MRI mapping technique. Results of this study validate this new MRI assessment by comparing its measurements to simultaneous water-perfusion manometry after ingestion of 240 ml of water with the aim to study pharmacokinetics in humans.

However, I have some remarks.

1. The aim of the study is a bit confusing. In the introduction, justification for this new technique is done by explaining the lack of methods for the assessment of predictive drug dissolution models in humans. This aim seems to change in the discussion and it highlights the use of this new method mainly and rather as a gastric motility assessment technique.

As the initial aim was to validate this new technique as a method to study pharmacokinetics and therefore dissolution of drugs, this should be further addressed more clearly in the discussion section. Additionally, a proof of concept study should be suggested as a next step to investigate the actual assessment of pharmacokinetics.

I do not agree with some section in the discussion addressing focus to the validity and accuracy of this new method for the assessment of gastric motility with a broader spectrum, including assessment of general GI function in GI disorder or the impact of drugs to GI motility. The ingestion of 240 ml of water studied in the present study is not enough to make such a statement. Therefore, careful should be taken in the discussion and it should be indicated for such cases further validation studies should be needed.

2. Where are the sensors of the water-perfusion manometry located? Is this antral and duodenal assessment? Please, specify in methods.

3. Following up the previous comment, if sensors are only located in the antral section of the stomach, and thus assessing antral contractions, this should also be specified throughout the entire manuscript: validation of this technique is for ANTRAL motility assessment and not general gastric motility assessment.

4. What was the drinking speed of the volunteers? Do they had to drink all water within 5 minutes or less? This was done in the MRI in supine position. How was this managed? Specify in methods.

5. In the methods it is discussed a “the manometer was connected to the MRI compatible water-perfused manometry system”. What does this means? That the manometry device is all-plastic? Is this the research’s group own developed system or was this bought from a company? In this last case, from which company?

6. Include reference and explain possible limitation regarding supine positioning and its effect on GI motility (Might have an effect on the gastric emptying rate): Treier R, Steingoetter A, Weishaupt D, Fried M, Boesiger P, Schwizer W. Gastroenterology 2003; Gastric motor function and emptying in the right decubitus and seated body position as assessed by magnetic resonance imaging

7. I think in the figures 4 and 5 the “AUC” for MRI is missing.

8. Figure 6. Can you include in the figure the moment the water was ingested? Why is there a gap between -15 and 0? Why is the pressure so low at time 0?

6. PLOS authors have the option to publish the peer review history of their article (what does this mean?). If published, this will include your full peer review and any attached files.

Reviewer #1: No

Reviewer #2: No

Reviewer #3: No

---

## [Author Response · Author response to Decision Letter 0]

25 Aug 2020

RESPONSE TO REVIEWERS ALSO UPLOADED AS WORD DOVUMENT

PONE-D-20-12517 REVISION R1: Measurement of fasted state gastric antral motility before and after a standard bioavailability and bioequivalence 240 mL drink of water: validation of MRI method against concomitant perfused manometry in healthy participants

We are grateful for the opportunity to revise our manuscript and we thank the Academic Editor and the Reviewers for their constructive comments. We believe that this process has considerably improved the paper. 

We address first the Journal/Editorial points raised and secondly the Reviewers’ comments in the point-by-point the detailed answer below. All changes made to the main manuscript are tracked in the marked-up Word version enclosed. Some Editorial points are also addressed in the enclosed Cover Letter as requested. 

Editorial and Journal points

Answer

Thank you, we have updated our disclosures as detailed below and in the Cover Letter. We have also resubmitted the figures after passing them through the PACE engine as required.

If applicable, we recommend that you deposit your laboratory protocols in protocols.io to enhance the reproducibility of your results. 

Answer

The study protocol is publicly available on Clinicaltrials.gov (NCT03191045) as already referenced in the manuscript. We do not have extensive protocols to deposit in protocols.io.

Answer

We ahve checked our manuscript against the templates adn checked the file naming, the format corresponds to the requirements

2. a) Please amend your current ethics statement to include the full name of the ethics committee/institutional review board(s) that approved your specific study.

2. b) Once you have amended this/these statement(s) in the Methods section of the manuscript, please add the same text to the “Ethics Statement” field of the submission form (via “Edit Submission”).

Answer

Thank you, the full name of the University of Nottingham Faculty of Medicine and Health Sciences Research Ethics Committee is now included at line 141. The full name of the FDA ethics committee already appeared. The statement now reads “The study was conducted at the Sir Peter Mansfield Imaging Centre (SPMIC) at the University of Nottingham. The study was approved by the University of Nottingham Faculty of Medicine and Health Sciences Research Ethics Committee (A14112016) and by the US Food and Drug Administration Research Involving Human Participants Committee (16-073D). All participants gave written, informed consent before joining the study.” and it has been added to the submission form on Editorial Manager.

3. We note that you have indicated that data from this study are available upon request. PLOS only allows data to be available upon request if there are legal or ethical restrictions on sharing data publicly. We will update your Data Availability statement on your behalf to reflect the information you provide.

Answer

Thank you, we had misunderstood this point. We have now saved the anonymized individual data and the time courses in an Excel file which we have uploaded to our Institution’s public repository, the Nottingham Research Data Management Repository (https://rdmc.nottingham.ac.uk/ ). 

The repository satisfies all major funders’ requirements for data access. The data file is now publicly available with this permanent DOI 10.17639/nott.7068 and this is noted also in the Cover Letter

4. Thank you for stating the following in the Financial Disclosure section:

' The authors GEA, GLA and LM received for this work a grant from the U.S. Food and Drug Administration (FDA), https://www.fda.gov/home, Contract HHSF223201510157C. The funders had no role in study design, data collection and analysis, decision to publish, or preparation of the manuscript. This article therefore reflects the views of the authors and should not be construed to represent FDA’s views or policies.'

We note that one or more of the authors are employed by a commercial company: Motilent Ltd, and Capsugel

a. Please provide an amended Funding Statement declaring these commercial affiliations, as well as a statement regarding the Role of Funders in your study. If the funding organization did not play a role in the study design, data collection and analysis, decision to publish, or preparation of the manuscript and only provided financial support in the form of authors' salaries and/or research materials, please review your statements relating to the author contributions, and ensure you have specifically and accurately indicated the role(s) that these authors had in your study. You can update author roles in the Author Contributions section of the online submission form.

Answer

Agreed, at point a. we have now included in the revised funding statement that co-author DMM is currently employed by company Capsugel and that co-author AM is CEO of Motilent Ltd and have specified in more detail their role in this work, also updated AM’s role more precisely in the ‘Author Contribution’ section. 

We also added the suggested statement at point on the role of the funder. 

REVISED FUNDING STATEMENT: ' The authors GEA, GLA and LM received for this work a grant from the U.S. Food and Drug Administration (FDA), https://www.fda.gov/home, Contract HHSF223201510157C. The author DMM is currently employed by company Capsugel and author AM is CEO of Motilent Ltd. The funders and company Capsugel had no role in study design, data collection and analysis, decision to publish, or preparation of the manuscript. Company Motilent provided the data analysis software as part of a subcontract on the research grant and had no role in study design, data collection and decision to publish. AM helped to with set up of the image analysis software and image data registration and with preparation of the final manuscript but had no role in the actual study data analysis. The funder provided support in the form of full salaries for authors KH, NA and fractional salary cost for CLH, JW GEA, GLA and LM, but did not have any additional role in the study design, data collection and analysis, decision to publish, or preparation of the manuscript. The specific roles of these authors are articulated in the ‘author contributions’ section.This article therefore reflects the views of the authors and should not be construed to represent FDA’s views or policies.'

b. Please also provide an updated Competing Interests Statement declaring these commercial affiliations along with any other relevant declarations relating to employment, consultancy, patents, products in development, or marketed products, etc. 

Within your Competing Interests Statement, please confirm that these commercial affiliations do not alter your adherence to all PLOS ONE policies on sharing data and materials by including the following statement: "This does not alter our adherence to PLOS ONE policies on sharing data and materials.” (as detailed online in our guide for authors http://journals.plos.org/plosone/s/competing-interests) . If this adherence statement is not accurate and there are restrictions on sharing of data and/or materials, please state these.

Please note that we cannot proceed with consideration of your article until this information has been declared.

Answer

Agreed. We have revised our Competing Interest statement accordingly as show below.

REVISED COMPETING INTEREST STATEMENT: I have read the journal's policy and the authors of this manuscript have the following competing interests: Authors GEA, GLA and LM were recipient of research grant # HHSF223201510157C by the U.S. Food and Drug Administration (FDA). This paid for research salaries and consumables for this work. The author AM is the CEO of Motilent Limited and the author DMM is an employee of Capsugel. This does not alter our adherence to PLOS ONE policies on sharing data and materials.

Answer

This has been included in our cover letter.

Additional changes

We have corrected affiliation 9 to the ‘Department of Statistics’ not Public Health as incorrectly stated

Response to Reviewer 1 Comments

“There is not much statistical content, other than some Pearson correlations and a mixed model. So I would argue about the term "spatio-temporal", as there is no multidimensional spatial/time series analyses, but only a simple repeated measures.”

Answer

Thank you for this comment. In the statistics literature, a spatio-temporal analysis models data both spatially and temporally. This study collected spatio-temporal data, but the planned analysis produced one number summaries of motility per subject and sampling period, thus collapsing the spatio-temporal data to two AUC statistics and losing some account of spatio-temporal qualities. As such we agree with the suggestion and have removed the “spatio-temporal” term throughout the manuscript unless specific, like for example in the case of the initial MRI raw data (STMM, reference 31) analysis technique which is published with the full spatio-temporal mapping features.

“Sample size is small and seems to have been chosen at random, as there is no power computation.”

Answer

Thank you for querying this, we agree a comment to this effect was missing. We have now included the following at Lines 255-266 (numbering from the clean version with no tracking): “The study design included power analysis considerations. Briefly, for correlation analyses involving data that are summarized to one number per subject, the standard error for the estimated correlation coefficient over 15 subjects is 0.28. However, our main analyses use time-resolved data with an overall sample size of 421 values over the 15 subjects. Using a Kronecker sum model to capture within-modality autocorrelation (for MRI or manometry) and between-modality correlation (the parameter of interest), much smaller standard errors are obtained. For example, if the within-modality autocorrelation is 0.5, the between-modality correlation coefficient is estimated with a standard error of 0.049. Thus, the confidence intervals for the correlation parameter of interest will be reported as the estimate +/- 0.1, giving us excellent power to resolve weak from strong correlation. The standard error depends on the within-modality autocorrelation, which was not known at the time of study design. The actual data obtained showed high precision of the estimated correlation coefficient, indicating that the a priori study design was successful."

“P-values on correlations are really meaningless, because they test 0 correlation against "some" correlation. In this case, a significant p-value doesn't necessarily mean the correlations are strong.”

Answer

Thank you for this, we agree with this point, we removed the p values on the correlations throughout. 

We also noted we had quoted R2 values which are now replaced by correlation r values

“The regression models need a careful residual analysis.”

Answer

Agreed, a careful residuals analysis was performed but not included in the manuscript. The residual analysis led us to conclude that the non-linear model fit the data better than the linear model.

“I think there could have been a lot more done with these data, but maybe the sample size is too small.”

Answer

The additional analysis was beyond the scope of the manuscript and what had been planned at funding stage. The possibility to use these data towards optimization and validation of in vitro and in silico predictive biopharmaceutic models will be explored. The sample size has now been addressed in answer to another point raised. 

Response to Reviewer 2 Comments

 “This manuscript reports on the use of non-invasive dynamic magnetic resonance imaging coupled with semi-automated spatio-temporal data analysis for measuring fasted-state gastric motility and validation of such a method vs. intraluminal pressure recording by means of perfused manometry technique.

 Human gastrointestinal motility patterns are of utmost importance to oral drug delivery. The investigated RMI-based method would represent a valuable advancement in the relevant study and allow to gain better insight into the impact of many variables. The work is very interesting, broad in scope, clearly exposed and in-depth discussed.”

Answer 

Thank you for the positive appraisal of our work.

“The title is rather long: is the word “imaging” actually needed when the acronym RMI already contains it?”

Answer

Agreed, we deleted the word ‘imaging’ and in response to Reviewer 1 we also deleted the ‘spatio-temporal mapping’ words thus making the title shorter.

“Lines 113-114: as the two methods correlated with each other, “both” sounds odd and can be deleted.”

Answer

Agreed, this was corrected as suggested.

“Lines 123-124: “Both studies did not use semi-automated analysis methods” may better read “Neither of the studies used…”.”

Answer 

Agreed, we rephrased this sentence as suggested.

“Line 125: please delete the comma after “making”.”

Answer

Agreed, the comma was deleted.

“Lines 127-128: “more operator independent” may be changed into “less operator-dependent” (this also applies to line 400).”

Answer 

Agreed, both sentences were adjusted as suggested.

“Line 143: “Exclusion criteria included..” sounds a bit strange. “Exclusion criteria were”? Or “encompassed”?”

Answer

Agreed, we used the word ‘encompassed’ instead of ‘included’.

“Fig. 2: the caption should be changed into a more concise form taking account of the detailed description and explanation already provided in the text. When the content of the image and the meaning of green and yellow lines are indicated, it will be enough.”

Answer

Agreed, we revised the figure legend of Figure 2 shortening it as follows: “Fig 2. Image analysis. Example of MRI data analysis. The stomach wall boundaries and the axis of the stomach are shown in green. The yellow lines represent the lumen diameter perpendicular to the stomach axis. AC, ascending colon; TC, transverse colon; GB, gall bladder.”

 “Line 257: “Each subject participated in either one or two intubation studies” seems to clash with the Methods section reporting that the volunteers “…took part in two identical fasted state MRI study visits” (lines 149-150) and “This was a single-centre, open-label design study that consisted of two separate identical study days...” (lines 154-155).”

Answer

Agreed, thank you, we deleted the sentence on line 257.

“Lines 297-298: please check and rephrase “The correlation for these data shows a significant positive correlation …”.”

Answer

Agreed, we rephrased the sentence: “A positive, significant correlation was observed between the two techniques (R² = 0.7134, p < 0.001).“

 “Fig. 6: please check the legend as colors are not visible in the figure whereas dotted/solid lines and different symbols are used as distinguishing features for the 2 curves.”

Answer 

Thank you for spotting this, this was a typo that slipped in from an older draft version, we now refer to the curves as solid and dotted lines. 

 “Lines 337-340: please check and rephrase” The fasting state and the use of a 240 mL dose of water as a challenge were specifically chosen because of their relevance to the bioavailability/bioequivalence (BA/BE) studies in the fasted state”, as it is implied that fasting volunteers take part in fasted-state bioavailability/bioequivalence studies. It seems that only 240 ml of water was chosen because of relevance to those studies.”

Answer

Thank you we checked the phrase and it is correct. During the clinical trials of oral drug product development, it is common practice (and FDA guidelines which we refer to) to ask the participants to fast overnight (hence the fasted state) and to take the drug product of interest drinking 240 mL of water. Our protocol copied this with the difference that our participant were not asked to ingest a drug product. 

This was not clearly stated so we have added now at lines Therefore, we added the following sentences at Lines 354-357: “However, it should be noted that the participants here did not actually ingest a drug product. The recommended 240 mL of water was applied as this volume is also used in BA/BE studies during the clinical phase of drug product development.” 

 “Lines 354-355: “When the stomach is active, this will cause the manometry AUC to increase more quickly than the MRI AUC….” does not appear strictly related to the previously-mentioned circumstance. Could this be an additional factor differing between the two methods?”

Answer 

We do see that the manometry AUC can increase more than the MRI AUC, but only seen for high pressure (mmHg) values. At a certain point, when the stomach completely contracts, no differences in MRI AUC will be observed but still some increase in manometry AUC can be noted as the pressure will increase (increasing force of the stomach towards the water-perfused electrodes), but distance will not change anymore. That is the reason why we see this non-linear correlation between both techniques at higher AUC values. The word ‘quickly’ however was vague and it has now been removed.

 “Line 364: should “complimentary” read “complementary”?”

Answer

Thank you the spelling was corrected.

 “Line 382: “a relative short time window” (or without “a”).”

Answer

Agreed we deleted “a”.

 “Line 393-394: “an oral administered” should read “an orally administered””

Answer

Thank you, this was adjusted.

“Line 396: as no statistical optimization study is concerned, “optimized” could be replaced by a different verb. The same applies to lines 418 and 438 (“optimization”).”

Answer

Thank you for this comment; we deleted the word ‘optimized’ and replaced it with ‘developed’ at Line 414, also deleted ‘optimization’ at old numbering Lines 418 and 436

 “Lines 315-319: this concept is already mentioned at lines 390-394. Please revise or synthesize the two similar passages in order to avoid repetitions.”

Answer

Having looked at this we think the Reviewer may have meant that lines 415-419 were repeating the concept at lines 390-394 about non-invasiveness, other parameters measurable and use for drug dissolution studies (315-319 are in the results relating to different things). We agree this was repeating itself a bit and we have now removed the lines at the second occurrence “The MRI technique is able to monitor different GI variables simultaneously (i.e. GI motility, gastric emptying and residual fluid volumes) which is of great interest for pharmaceutical purposes in order to study the impact of GI variables on drug and formulation behaviour.”

 “Lines 431: “gastric motility” can be deleted as it is specified again at line 432, same sentence.”

Answer

Agreed, “Gastric motility” was deleted at the first occurrence.

Response to Reviewer 3 Comments

 “The manuscript is interesting and well written. It is not the first time that MRI has been shown to be a valid method for the measurement of gastric emptying, secretion, motility and intragastric distribution of gastric contents. This study, however, adds value to the use of MRI that assess gastric motility by using a new spatio-temporal MRI mapping technique. Results of this study validate this new MRI assessment by comparing its measurements to simultaneous water-perfusion manometry after ingestion of 240 ml of water with the aim to study pharmacokinetics in humans.”

Answer

Thank you for this positive appraisal.

 “However, I have some remarks.

1. The aim of the study is a bit confusing. In the introduction, justification for this new technique is done by explaining the lack of methods for the assessment of predictive drug dissolution models in humans. This aim seems to change in the discussion and it highlights the use of this new method mainly and rather as a gastric motility assessment technique. As the initial aim was to validate this new technique as a method to study pharmacokinetics and therefore dissolution of drugs, this should be further addressed more clearly in the discussion section. Additionally, a proof of concept study should be suggested as a next step to investigate the actual assessment of pharmacokinetics.”

Answer

Correct statement. During the past years, there was specific focus on validating existing in vitro and in silico models for their capacity to predict the in vivo performance of an orally administered drug product to humans. Numerous projects (e.g., OrBiTo, PEARRL, UNGAP) were designed to explore the predictive performance of existing in vitro and in silico models. However, there is still a general scarcity of data on gastric motility to help us understand the impact of gastric motility on an orally administered dosage form. With the MRI techniques one can look at undisturbed (no intubation) contractility. As a recent study by Hens et al. successfully implemented MRI-derived dynamic fluid volumes in commercially-available computational software (e.g., GastroPlus), we will be able now to do the same for motility patterns and investigate how this will impact formulation disintegration for instance. 

We agree with you that something was missing in the Discussion section. We added there at Lines XXX-YYY a sentence to better link this study with pharmacokinetic studies: “With respect to future uses of the MRI technique, a pharmacokinetic study in combination with assessment of motility could provide unprecedented insights on inter-subject variability in systemic outcome of a drug, as the underlying motility patterns are likely one dominant cause of known inter-subject differences in formulation disintegration and, subsequently, dissolution and absorption of the drug. These type of studies are the next logical step to further our understanding of to investigate the actual assessment of pharmacokinetics variables”

 “I do not agree with some section in the discussion addressing focus to the validity and accuracy of this new method for the assessment of gastric motility with a broader spectrum, including assessment of general GI function in GI disorder or the impact of drugs to GI motility. The ingestion of 240 ml of water studied in the present study is not enough to make such a statement. Therefore, careful should be taken in the discussion and it should be indicated for such cases further validation studies should be needed.”

Answer

Agreed, the last 5 lines of the Discussion were speculative as to the validity and accuracy of these methods beyond the data presented and were removed. The corresponding statements in the Abstract and in Conclusions have been removed accordingly.

 “2. Where are the sensors of the water-perfusion manometry located? Is this antral and duodenal assessment? Please, specify in methods.”

Answer

Good point, agreed, we have clarified at Lines 180-182 that “The perfused manometry ports tend to record primarily antral contractions whilst more proximal sensors in the wider fundal region do not record contractions well, as such the assessment here was focused on the antral region of the stomach.”

 “ 3. Following up the previous comment, if sensors are only located in the antral section of the stomach, and thus assessing antral contractions, this should also be specified throughout the entire manuscript: validation of this technique is for ANTRAL motility assessment and not general gastric motility assessment.”

Answer

Agreed, we have specified ‘antral motility’ in the title and several times throughout the manuscript.

 “4. What was the drinking speed of the volunteers? Do they had to drink all water within 5 minutes or less? This was done in the MRI in supine position. How was this managed? Specify in methods.”

Answer

Agreed, we now specify at Lines 206-208 that the participants drank while they were sitting up on the scanner bed and that they were not instructed to drink the water at a given speed, but all of the participants drank the glass of water quickly, within approximately one minute.

 “5. In the methods it is discussed a “the manometer was connected to the MRI compatible water-perfused manometry system”. What does this means? That the manometry device is all-plastic? Is this the research’s group own developed system or was this bought from a company? In this last case, from which company?”

Answer

Agreed, some detail was missing, thank you for pointing this out, we now added at lines 183-187 the system manufacturer (the Biomedical Engineering Department at The Royal Melbourne Hospital in Australia) and describe that the transducers connecting to the catheter were secured to a MRI-compatible trolley placed next to the scanner bed. Electric cables and pipes went through a wave guide opening and connected the transducers box to the electronics and pressure gas cylinder placed outside the scanner room. 

 “6. Include reference and explain possible limitation regarding supine positioning and its effect on GI motility (Might have an effect on the gastric emptying rate): Treier R, Steingoetter A, Weishaupt D, Fried M, Boesiger P, Schwizer W. Gastroenterology 2003; Gastric motor function and emptying in the right decubitus and seated body position as assessed by magnetic resonance imaging”

Answer

Thank you for raising this point. We added a comment with reference to that paper and also to the other paper from the same group whereby they put people upside down in a vertical scanner at lines 438-443 with the two references. 

 “I think in the figures 4 and 5 the “AUC” for MRI is missing.”

Answer

Thank you for spotting this, this was added to both figures.

 “8. Figure 6. Can you include in the figure the moment the water was ingested? 

Answer

Agreed, the drink ingestion time has been indicated with an arrow and text in Fig 6.

Why is there a gap between -15 and 0? 

Answer

As described at Lines 207-208 the participants were allowed a rest outside the scanner room before the drink part of the study. To allow them to move we disconnected the perfused manometry system (Line 208). None of the participants wanted a long break, they took only advantage of the rest time to stretch their legs and visit the toilet before continuing and the average time including reconnection was 15 minutes corresponding to the gap. We have clarified the timing at liens 208-211

Why is the pressure so low at time 0?”

Answer

We have a comment to this effect at Lines 365-368 of the Discussion where we indicate that is likely to reflect the inability of the perfused manometry to measure non-occlusive events when a free fluid is present to redistribute the pressure changes instantaneously. 

Figures Check

Answer

Agreed, we have passed the figures through the PACE engine and uploaded the PACE Corrected files

---

## [Decision Letter · Decision Letter 1]

29 Sep 2020

PONE-D-20-12517R1

Measurement of fasted state gastric antral motility before and after a standard bioavailability and bioequivalence 240 mL drink of water: validation of MRI method against concomitant perfused manometry in healthy participants

PLOS ONE

Dear Dr. Marciani,

Thank you for submitting your manuscript to PLOS ONE. After careful consideration, we feel that it has merit but does not fully meet PLOS ONE’s publication criteria as it currently stands. Therefore, we invite you to submit a revised version of the manuscript that addresses the points raised during the review process.

We look forward to receiving your revised manuscript.

Kind regards,

Florencia Carbone

Academic Editor

PLOS ONE

Additional Editor Comments (if provided):

After carefully reviewing the manuscript no main issues were found.

However, I only have two comments that might need some work:

1. It is consented that for the assessment of antral motility, the frequency and amplitude of contractile waves are measured in order to define a motility index (See reference: PMID: 19019032). The motility index (MI) represents the fraction of time during which gastric motility was detected and is normally calculated as an average of the individual detected contractions in a specific time window taking into account the relative amplitude of each contractile wave. Motility index is then a composite parameter that incorporates both contraction frequency and amplitude. May I suggest, in order to add value and credibility to your study, to think about incorporating the term of MI in your analysis. MI of both techniques will also be easier to correlate. I think this is in some way what you are doing by assessing AUCs, but in this way, the acquired information with MI will be cleaner and easier to understand for the reader.

2. Methods regarding the intragastric manometry are a bit vague. From the manuscript it seems as the original method to assess intragastric pressure has been adapted to be used in a MRI setting. Do I understand correctly that the total length of the manometry tube was 280 cm (180+100 pigtails) in order to allow the pressure to be measured from outside the MRI room? This is quite a distance. As the water-perfusion manometer is normally not validated to acquire pressure information at such a distance, how was the pressure on the tip of the manometer (water drop) double-checked to be correct? Have you performed any validation of this method to verify accurate assessment of values? This is very important as you are using this data to validate another technique (the MRI). You should include this information on your methods.

Reviewers' comments:

Reviewer's Responses to Questions

**Comments to the Author**

1. If the authors have adequately addressed your comments raised in a previous round of review and you feel that this manuscript is now acceptable for publication, you may indicate that here to bypass the “Comments to the Author” section, enter your conflict of interest statement in the “Confidential to Editor” section, and submit your "Accept" recommendation.

Reviewer #1: All comments have been addressed

2. Is the manuscript technically sound, and do the data support the conclusions?

Reviewer #1: (No Response)

3. Has the statistical analysis been performed appropriately and rigorously? 

Reviewer #1: (No Response)

4. Have the authors made all data underlying the findings in their manuscript fully available?

Reviewer #1: (No Response)

5. Is the manuscript presented in an intelligible fashion and written in standard English?

Reviewer #1: (No Response)

6. Review Comments to the Author

Reviewer #1: (No Response)

7. PLOS authors have the option to publish the peer review history of their article (what does this mean?). If published, this will include your full peer review and any attached files.

Reviewer #1: No

---

## [Author Response · Author response to Decision Letter 1]

12 Oct 2020

We are grateful for Reviewer #1’s acknowledgement that we have addressed all the comments from the first round of reviews, and for the Additional Editor’s appraisal that the manuscript does not have any main issues. We welcome the opportunity to answer the two remaining Additional Editor’s comments as detailed below.

All changes made to the main manuscript version R2 are tracked in the marked-up Word version enclosed and a separate clean unmarked version is also uploaded. 

1. It is consented that for the assessment of antral motility, the frequency and amplitude of contractile waves are measured in order to define a motility index (See reference: PMID: 19019032). The motility index (MI) represents the fraction of time during which gastric motility was detected and is normally calculated as an average of the individual detected contractions in a specific time window taking into account the relative amplitude of each contractile wave. Motility index is then a composite parameter that incorporates both contraction frequency and amplitude. May I suggest, in order to add value and credibility to your study, to think about incorporating the term of MI in your analysis. MI of both techniques will also be easier to correlate. I think this is in some way what you are doing by assessing AUCs, but in this way, the acquired information with MI will be cleaner and easier to understand for the reader.

Answer: Thank you for this comment. We agree with you that the motility index is a common and accepted parameter in the field of gastrointestinal motility, indeed one of our co-authors here is also a co-author of the consensus statement that you reference. For our study, after much deliberation, we had chosen to use the area under the curve (AUC), which is another one of the ‘classic’ contratile parameters used to analyze motility manometry traces. The AUC incorporates both the strength and frequency of contractions in an integral fashion providing a single non-composite number. The AUC outcome was then stated on the contract with the FDA and declared as primary outcome of the study on the protcol registration. For the above reasons we would very respectfully ask to retain the AUC data analysis. We have however added a comment at lines 432-435 to acknowledge the possible limitation and consider future use of the motility index. 

2. Methods regarding the intragastric manometry are a bit vague. From the manuscript it seems as the original method to assess intragastric pressure has been adapted to be used in a MRI setting. Do I understand correctly that the total length of the manometry tube was 280 cm (180+100 pigtails) in order to allow the pressure to be measured from outside the MRI room? This is quite a distance. As the water-perfusion manometer is normally not validated to acquire pressure information at such a distance, how was the pressure on the tip of the manometer (water drop) double-checked to be correct? Have you performed any validation of this method to verify accurate assessment of values? This is very important as you are using this data to validate another technique (the MRI). You should include this information on your methods.

Answer: Thank you this comment, we agree some details were unclear. We have now clarified at lines 184 – 191 that the pressure was transduced to electrical signal inside the scanner room, right at the bore of the scanner. We also clarify that we used the old manual, gravity method to calibrate the manometry system, before and after the study, with the claibrations recorded on the traces so that we could detect drifts or malfunctions. We agree that the total 280 cm from tip to end pigtail is long, this was to allow for the catheter to reach comfortably the transducers from inside the bore of the scanner. One typical concern with long catheters is lag of response but the system was responding rapidly to gravity changes, with no visible lag. The same system had been used and published by our colleagues in Zurich.

---

## [Editor Report · Decision Letter 2]

15 Oct 2020

Measurement of fasted state gastric antral motility before and after a standard bioavailability and bioequivalence 240 mL drink of water: validation of MRI method against concomitant perfused manometry in healthy participants

PONE-D-20-12517R2

Dear Dr.,

We’re pleased to inform you that your manuscript has been judged scientifically suitable for publication and will be formally accepted for publication once it meets all outstanding technical requirements.

Kind regards,

Florencia Carbone

Guest Editor

PLOS ONE

Additional Editor Comments (optional):

Reviewers' comments:

Thank you for the clarifications. Everything is very clear now.

---

## [Editor Report · Acceptance letter]

21 Oct 2020

PONE-D-20-12517R2 

Measurement of fasted state gastric antral motility before and after a standard bioavailability and bioequivalence 240 mL drink of water: validation of MRI method against concomitant perfused manometry in healthy participants  

Dear Dr. Marciani:

I'm pleased to inform you that your manuscript has been deemed suitable for publication in PLOS ONE. Congratulations! Your manuscript is now with our production department. 

Kind regards, 

on behalf of

Dr. Florencia Carbone 

Guest Editor

PLOS ONE